# Evaluation of Two Different CMV-Immunoglobulin Regimens for Combined CMV Prophylaxis in High-Risk Patients following Lung Transplant

**DOI:** 10.3390/microorganisms11010032

**Published:** 2022-12-22

**Authors:** Víctor M. Mora, Piedad Ussetti, Alicia de Pablo, David Iturbe, Rosalía Laporta, Rodrigo Alonso, Myriam Aguilar, Carlos A. Quezada, José M. Cifrián

**Affiliations:** 1Service of Respiratory Medicine, Hospital Universitario Marqués de Valdecilla, 39008 Santander, Spain; 2Genetic Epidemiology and Atherosclerosis in Systemic Inflammatory Diseases Group, IDIVAL, 39008 Santander, Spain; 3Department of Neumology, Hospital Puerta de Hierro, Majadahonda, 28222 Madrid, Spain; 4Department of Neumology, Hospital Universitario 12 de octubre, 28041 Madrid, Spain

**Keywords:** cytomegalovirus, lung transplantation, CMV infection, prophylaxis, CMV immunoglobulin, morbidity, survival

## Abstract

Background: The clinical benefits of the common off-label use of cytomegalovirus (CMV)-specific immunoglobulin (CMV-Ig) combined with antivirals in organ transplantation have not been previously assessed. The objective was to compare the risk of CMV infection and other post-transplantation outcomes between two CMV-Ig prophylaxis regimens in lung transplant recipients; Methods: Retrospective study of 124 donor CMV positive/recipient negative (D+/R–) patients receiving preventive ganciclovir/valganciclovir for 12 months, of whom 62 received adjunctive CMV-Ig as per label indication (short regimen [SR-Ig]; i.e., 7 doses over 2.5 months) and were compared to 62 who received an extended off-label regimen (ER-Ig) consisting of 17 doses over one year after transplantation. Results: The incidence of CMV infection or disease, acute rejection, chronic lung allograft dysfunction, and survival did not differ between the two CMV-Ig schedules. Although the time to the first CMV infection after transplantation was shorter in the ER-Ig than in the SR-Ig adjunctive group (log-rank: *p* = 0.002), the risk was independently predicted by antiviral cessation (odds ratio = 3.74; 95% confidence interval = 1.04–13.51; *p* = 0.030), whereas the CMV-Ig schedule had no effect. Conclusions: Extending the adjunctive CMV-Ig prophylaxis beyond the manufacturer’s recommendations up to one year does not confer additional clinical benefits regarding lung post-transplantation outcomes.

## 1. Introduction

Infections are the leading cause of death between 30 days and one year after lung transplantation [1]. Cytomegalovirus (CMV) is largely documented as one of the most important opportunistic viral pathogens [2], with both CMV infection and disease (defined as CMV infection with attributable symptoms) [3] reported in approximately 40% of those receiving lung and heart-lung transplant [4].

CMV infection usually occurs during the first year after transplantation [5], ranging from asymptomatic to viral syndrome to tissue-invasive disease (e.g., colitis, gastritis, hepatitis, nephritis, pneumonitis, retinitis, etc.). In addition, indirect effects can lead to severe complications such as other secondary opportunistic infections, post-transplantation lymphoproliferative disorder (PTLD), graft dysfunction and failure, acute rejection, transplant and nontransplant vascular disease, new-onset diabetes, or bronchiolitis obliterans syndrome (BOS) [5]. Besides the clinical impact, CMV infection increases healthcare resource use and hospitalization costs and negatively impacts the patient’s quality of life compared with those without CMV disease [6,7].

The risk of developing CMV disease depends mainly on pre-transplant serology, with the highest risk among seronegative subjects who receive (R) an organ from a seropositive donor (D; D+/R– or CMV mismatch) [8,9]. This is because of the reactivation of latent virus transmitted in the allograft to which the recipient cannot respond effectively due to the lack of pre-existing CMV-specific cell-mediated immunity and the pharmacological immunosuppression needed to avoid allograft rejection [5]. Data from the latest ISHLT (International Society for Heart and Lung Transplantation) registry showed that 44% of subjects who received a lung transplant were CMV antibody negative and identified CMV mismatch as an independent predictor of 1- and 5-year mortality [10].

According to national and international guidelines, prevention of CMV is considered the standard of care for all individuals at risk receiving a lung transplant [8,9,11], as it prevents CMV disease in 58–80% of cases [12]. The two main preventive strategies are universal antiviral prophylaxis and pre-emptive therapy [8,9,11]. Universal prophylaxis involves administering antiviral agents, often at half the treatment dose, starting shortly after transplantation and continuing for 6–12 months. The most commonly used drugs are valganciclovir and ganciclovir (oral or intravenous). In addition to antiviral prophylaxis, adjunctive CMV-specific immunoglobulin (CMV-Ig) or intravenous immunoglobulin (IV-Ig) is an option in high-risk lung and heart transplant recipients. Pre-emptive therapy consists of weekly or biweekly immunological monitoring for the first few months after transplantation and standard treatment dose antiviral therapy in patients with a positive assay. The incidence of CMV disease within 24 months among lung transplant recipients who received universal prophylaxis was estimated to be 19.4% [6,13], with a higher cumulative probability of developing the disease in D+/R– pairs (33.7% vs. 8.9% in D+/R+ and 0% in D–/R+ recipients) [13].

The utility of CMV-Ig has been debated after conflicting results reported in different meta-analyses evaluating its effectiveness following solid organ transplantation (SOT): An early meta-analysis concluded that the risk of CMV infection, CMV disease, CMV invasive disease, risk of acute rejection, graft loss, opportunistic infections, and all-cause mortality was not different between antiviral medication combined with CMV-Ig vs. only antiviral prophylaxis [14]. However, a further meta-analysis including only randomized clinical trials found that prophylactic use of CMV-Ig was associated with increased survival, decreased risk of CMV-associated death and CMV disease but had no effect on CMV infections and clinically relevant rejections compared to placebo, no treatment, or antiviral prophylaxis alone [15]. Lastly, the most recent systematic review and meta-analysis reported that the rate of CMV infection was significantly lower among those receiving CMV-Ig prophylaxis (35.8% vs. 41.4%) with no difference in the time to infection compared with no CMV-Ig prophylaxis, a non-CMV-Ig prophylactic treatment, or placebo [16].

The lack of evidence-based guidance supporting the use of CMV-Ig results in heterogenous routine clinical practice [5]. International surveys on the management of CMV in lung transplantation in D+/R– patients have reported that CMV-Ig is used as part of the universal prophylaxis strategy by 32% of centers and 38% of clinicians [17,18]. Moreover, a recent systematic review on the effectiveness of CMV-Ig adjunctive prophylaxis following SOT reported that the dosing regimen and schedule were highly variable between studies and centers [16]: Among those using CMV-Ig in lung transplants, the most common regimen was 150 mg/kg within 72 h of transplantation and every 2 weeks thereafter, but the dosing ranged between 100 and 150 mg/kg and 1–2 mL/kg in some studies. Additionally, the number of doses ranged between 1 and 12, the interval between doses between 1 day and 1 month, and the median duration of therapy post-transplantation between 1 month and 1 year [16]. The authors concluded that there is a lack of real-world non-interventional studies assessing the diverse off-label use of CMV-Ig in routine clinical practice, which would provide important insights into the clinical benefits of CMV-Ig [16].

As with international clinical guidelines, Spanish experts and consensus documents do not provide guidance on passive CMV prophylaxis with CMV-Ig in lung transplantation [11,19]. Therefore, some centers have implemented its adjunctive use with antivirals mainly based on each center’s experience and protocols. The present retrospective study aimed to compare the risk of CMV infection and other post-transplantation outcomes between two adjunctive CMV-Ig prophylaxis (i.e., different dose, dosing interval, and therapy duration) in high-risk CMV mismatched (D+/R–) lung transplant recipients from three separate centers in Spain.

## 2. Materials and Methods

### 2.1. Study Design and Patients

Three hospitals in Spain, which are accredited members of the National Transplant Organization (ONT), conducted a multicenter, retrospective, non-interventional study using electronic medical records of adult patients undergoing lung transplantation between 1 January 2009, and 31 December 2020. The study included pretransplant seronegative patients (defined as negative CMV IgG; R−) receiving a CMV seropositive allograft (defined as CMV IgG positive donor; D+) who received universal antiviral prophylaxis with adjunctive CMV-Ig.

Ethical approval for this study was obtained from the Research Ethics Committees for Medicines and Medical Devices (CEIm) of Cantabria, Spain (code 2022.227). The study was conducted in accordance with the Declaration of Helsinki (2013).

### 2.2. Immunosuppressive Protocol

Induction of immunosuppression was the same in all participating centers and included basiliximab (20 mg intravenously [IV] on day 0 and day 4), except that one of the centers only used basiliximab in selected patients (>65 years, severe pulmonary hypertension, or renal failure) before 1 April 2016. Additionally, all centers used a triple maintenance immunosuppression protocol consisting of a calcineurin inhibitor (CNI; tacrolimus or cyclosporine), an anti-metabolite agent (mycophenolate mofetil or azathioprine), and a corticosteroid (prednisolone).

### 2.3. CMV Prophylaxis

#### 2.3.1. Antiviral Regimen

All participant centers used the same protocol for D+/R– pairs, namely ganciclovir (GC, at 5 mg/kg/day) during the first week post-transplantation followed by oral valganciclovir (VGCV; 900 mg/day) for 12 months, with the doses adjusted for renal impairment according to the manufacturer’s recommendations.

#### 2.3.2. Adjunctive CMV-Ig Regimens

All participant centers used the same IV human CMV IgG preparation (Megalotec^®^, formerly Cytotect^®^, Biotest Pharma GmbH, Dreieich, Germany). Patients were classified into two groups based on the CMV-Ig prophylaxis regimen of the institutional protocol:
Label use or short regimen (SR-Ig) regimen, used in one center and given according to the Summary of Product Characteristics (SmPC), namely one 150 IU/kg dose on the day of the transplant, then six 100 IU/kg additional doses given at 2, 7, 14, 21, 35, 56, and 77 days post-transplantation [20].Off-label dosage or extended regimen (ER-Ig), used in the other 2 hospitals and consisting of 2 mL/kg (200 UI/kg) on days 1, 4, 8, 15, and 30 post-transplant, then monthly for 1 year thereafter [21].


### 2.4. Variables and Outcomes Assessed 

Data collected retrospectively from electronic medical records included age and sex; CMV serologic status of donor and recipient; age at transplantation; type of transplant (unilateral or bilateral); indication for transplantation; immunosuppression induction (yes/no); immunosuppressive drugs used; duration of GCV/VGCV prophylaxis; use of antiviral prophylaxis other than GCV/VGCV; and CMV-Ig prophylaxis regimen group (SR-Ig or ER-Ig).

Post-transplantation outcomes measures included duration of antiviral prophylaxis, premature discontinuation (yes/no) and reason for discontinuation; presence, number, and severity grade of acute rejection (AR) episodes based on definitions proposed by the ISHLT [22], number of AR episodes treated with IV megadose or oral corticoids; occurrence of CMV primoinfection (yes/no; defined as the presence of viremia in peripheral blood requiring specific anti-CMV treatment) with date if applicable; presence and organ affected if CMV disease in primoinfection (defined as CMV infection with proven organ damage); number of post-transplantation CMV infections and whether they required oral or IV treatment; and chronic lung allograft dysfunction (CLAD; yes/no) with initiation date and type of CLAD, namely bronchiolitis obliterans syndrome (BOS) and restrictive allograft syndrome (RAS), all based on definitions proposed by the ISHLT when applicable [23,24]. Finally, the time to follow-up or date and cause of death in patients not surviving was also extracted.

### 2.5. Statistical Analysis

The analytical populations included patients undergoing CMV-Ig prophylaxis with the SR-Ig schedule and those on the alternative ER-Ig approach. Descriptive statistics were used to summarize the patient characteristics for each group, with continuous variables shown as mean and standard deviation (SD) or as median and interquartile range (IQR) and categorical variables as number (percentage). For continuous variables, the crude difference between groups was assessed by Student’s *t*-test or Mann-Whitney U test, and to compare proportions, the Chi-square test. A Kaplan-Meier analysis was used to evaluate the time to first CMV infection, time to first CMV infection off prophylaxis, time to CLAD, and survival rate. The log-rank test was applied to evaluate differences between groups starting from the day of the transplantation. A logistic regression analysis was conducted to determine factors associated with CMV infection and was expressed as odds ratio (OR) and 95% confidence interval (CI). The risk of the first CMV infection since the discontinuation of the antiviral prophylaxis was assessed by the Kaplan-Meier estimator and a Cox regression analysis expressed as hazard ratio (HR) and 95% CI. Both analyses were adjusted for the effect of significant variables in the univariate analyses. All hypothesis contrasts were bidirectional, and the statistical significance level was set at 0.05. Data documentation and statistical analyses were performed using SPSS version 20.0 (SPSS Inc., Chicago, IL, USA).

## 3. Results

During the study period, 1280 adult patients underwent a lung transplant in the three participant centers, of whom 124 were high-risk CMV D+/R– pairs. In two of the centers, a total of 62 recipients (39 recruited in one center and 23 in the other) received the same 12-month extended CMV-Ig adjunctive prophylaxis (ER-Ig) schedule, and in the other 62 recipients received the CMV-Ig as per label indications (SR-Ig; 2.6 months). Overall, the patients were followed for a median of 3.7 years (IQR, 2.0–6.9).

Recipients ranged in age between 38.0 and 60.8 years (median = 51.6), and half were male (Table 1). Most transplantations were bilateral (83.1%), and the most frequent indication was interstitial lung disease (ILD; 32.3%), followed by bronchiectasis (24.2%) and chronic obstructive pulmonary disease (COPD; 23.4%). 

Induction therapy with basiliximab was administered in 81.5% of patients and was more frequently used in centers using the ER-Ig than in the one treating with the SR-Ig regimen (100% vs. 62.9%; *p* < 0.001). Regarding maintenance therapy, there were no significant differences between the two CMV-Ig schedule groups except for a more frequent use of azathioprine and mTOR inhibitors in patients with the ER-Ig regimen (*p* < 0.001; Table 1).

### 3.1. Antiviral CMV Prophylaxis

VGCV was given for a median of 12 months after transplantation in both groups, but the range was significantly shorter in the ER-Ig schedule (IQR = 7–12 vs. 12–13; *p* = 0.003) (Table 2). Overall, VGCV was prematurely discontinued in 31 individuals (25.0%) and cessation was more frequent in recipients of the ER-Ig group (26 vs. 5 subjects; *p* < 0.001) (Table 2). Myelotoxicity and renal failure were the most frequent reasons in the ER-Ig group (12 and 7 patients, respectively). Although CMV breakthrough was the main reason for premature discontinuation in the SR-Ig group (3 out of 5 discontinuations), the total number of discontinuations due to CMV infection in the ER-Ig group was similar (4 patients). The median time of VGCV treatment was 7 months (IQR = 5–10) among those who discontinued and 12 months (IQR = 12–13) among those who did continue (*p* < 0.001).

### 3.2. Incidence and Type of CMV Episodes

Overall, 74.0% of lung recipients developed a CMV infection, with no significant differences between CMV-Ig schedules (67.7% in the SR-Ig and 80.6% in the ER-Ig group, respectively) (Table 2).

The first CMV infection after the transplantation occurred later in the SR-Ig than in the ER-Ig group (median = 14.8 vs. 11.1 months; *p* < 0.001) (Table 2). The Kaplan-Meyer survival analysis showed that the time to first CMV infection after transplantation was longer in the SR-Ig adjunctive prophylaxis schedule (log-rank: *p* = 0.002; Figure 1a). The one-year cumulative proportion of patients free from CMV primoinfection was 91.4% in the SR-Ig group and 50.9% in the ER-Ig group and the ER-Ig group, respectively (log-rank *p* = 0.001).

The time to the first CMV infection from the end of the antiviral prophylaxis was not statistically different between schedules (median = 8.6 vs. 6.9 weeks; log-rank *p* = 0.805; Figure 1b). To explore the effect of the VGCV discontinuation in the time to the first CMV infection, we conducted additional Kaplan-Meyer analyses: The time did not differ between those who discontinued prematurely and those who did not (log-rank *p* = 0.224; Appendix A) and was also similar between CMV-Ig schedules among the subgroup of recipients who completed the 12 months of VGCV prophylaxis (log-rank *p* = 0.336; Appendix A).

The frequency of CMV disease during the first CMV infection did not differ between CMV-Ig regimens (11.9% and 12% in the SR-Ig and ER-Ig group, respectively), nor was the affected organ (Table 2).

The total number of CMV infections during follow-up was significantly higher among ER-Ig than SR-users (range 1–2 vs. 1–3; *p* < 0.001), with no difference in the frequency between those treated with IV or oral treatment between groups. (Table 2).

### 3.3. Risk Factors for CMV Infection

The univariate logistic regression analysis showed that significant factors associated with a shorter time to first CMV infection were lower age at transplant (*p* = 0.034) and premature antiviral prophylaxis discontinuation (*p* = 0.026; Table 3). By multivariate analysis, only premature discontinuation of antiviral prophylaxis remained significant (OR = 3.74; 95% CI = 1.04–13.51; *p* = 0.030; Table 3).

The only significant risk factor for the time to first CMV infection off antiviral prophylaxis by univariate analysis was pulmonary arterial hypertension indication for transplant (PAH; *p* = 0.005), which remained significant by multivariate analysis, with four times higher risk in patients with PAH than in those without (HR = 4.16; 95% CI = 1.53–11.3; *p* = 0.005; Appendix A).

The Kaplan-Meyer analysis for the cumulative risk of the first CMV infection off antiviral prophylaxis showed no differences between the two CMV-Ig schedules (log-rank *p* = 0.441; Appendix A).

### 3.4. Incidence of Acute Cellular and Chronic Allograft Rejection

The number of ARs was similar between CMV-Ig prophylaxis groups (IQR = 1–2 for both schedules), and there was no difference in the severity grading (Table 2). Finally, the number of AR treated with megadose of corticosteroids or oral corticosteroids was also comparable between groups.

CLAD was reported in a similar proportion of recipients in each group (24.2% and 27.4% in the SR-Ig and ER-Ig, respectively; Table 2). No significant difference was found in freedom from CLAD (log-rank: *p* = 0.846), which was 96.4% in the SR-Ig group and 98.3% in the ER-Ig group at 1-year of the transplant and 15% in the SR-Ig group and 22.2% in the ER-Ig group after 3 years (Figure 1C). BOS was the most frequent CLAD phenotype (90.6% of all cases), and there were no significant differences in the occurrence of BOS and RAS between CMV-Ig prophylaxis groups. Although not statistically significant, the time from the transplant to CLAD was slightly longer among recipients treated with the SR-Ig approach (median 3.18 vs. 3.15 years; *p* = 0.055).

### 3.5. Survival

Forty-one patients (33.1%) died during the follow-up, 21 in the SR-Ig group and 20 in the ER-Ig group (*p* = 0.849; Table 2). Only one patient, in the SR-Ig schedule, died as a direct result of the CMV disease. The cumulative proportion of patients surviving after the transplant was not different between CMV-Ig schedules (log-rank: *p* = 0.345; Figure 1D), with 91.9% and 95.2% of patients alive after 1 year in the SR-Ig and ER-Ig groups, respectively.

## 4. Discussion

In this real-world, retrospective study, we assessed two adjunctive CMV-Ig prophylaxis schedules (2.6 vs. 12 months duration) in D+/R–lung recipients receiving GCV/VGCV as antiviral preventive agents. The results showed that the CMV infection or disease incidence, AR, CLAD, and survival did not differ between the two CMV-Ig schedules. Although the time to the first CMV infection after transplantation was shorter in the extended (ER-Ig) than in the label use (SR-Ig) adjunctive prophylaxis, the risk was independently predicted by antiviral cessation but not the CMV-Ig schedule group. This suggests that a longer rather than shorter CMV-Ig regimen does not confer any real advantage on the assessed post-transplantation outcomes.

The characteristics of the lung recipients assessed in our study align with those reported by the most recent ISHLT reports [10,25]. Namely, the median age was 51.6 years, almost 60% of the patients were male, and 83.1% had bilateral procedures [10,25].

In our study, 74% of patients developed a CMV infection and 12% CMV disease, with no significant differences between CMV-Ig schedules. These figures are in line with those reported by available studies using combined GCV/VGCV and either long (12 months) or short (1 month) CMV-Ig prophylaxis schedules in adult lung recipients, namely 61–73% of CMV infections [21,26] and 13.2–15.6% of CMV disease [21,27,28]. There are few available studies assessing different CMV-Ig dosing (i.e., length of prophylaxis) in thoracic transplant recipients. A retrospective review of pediatric lung recipients receiving GCV and CMV-Ig for a mean of 5 doses and a median dose of 150 mg/kg, reported a lower risk of developing CMV infection 1 year after transplantation compared with no CMV-Ig prophylaxis [29]. Although this could indicate that only 1 month of CMV-Ig treatment would be efficacious, there was a lack of uniformity in the schedule and dosage between sites (in Europe and the US), with doses ranging between 1 and 12 and dosing intervals between 1 day and 1 month, which precluded definitive conclusions. Additionally, a retrospective study in D+/R− heart transplant patients reported that those receiving antiviral prevention and 1 dose of IV nonspecific immunoglobulin therapy (500 mg/kg) followed by one dose of CMV-Ig (125 mg/kg) had higher rates of CMV disease after 2 years of transplant than those who received 5 doses of CMV-Ig given over 2 months (once weekly for 4 weeks followed by two doses 2 weeks apart, a schedule similar to that used in the SR-Ig regimen in our study) [30].

The only significant differences between the short and extended CMI-Ig regimens that we observed were a significantly shorter time to first CMV infection and a higher number of CMV infections among those in the ER-Ig than the SR-Ig schedule. However, these differences were most probably attributable to the higher proportion of recipients in the ER-Ig group who discontinued prematurely the VGCV therapy (41.9% vs. 8.1%). Several findings in our study support this hypothesis. Firstly, the median time to CMV infection off VGCV prophylaxis—which was 7.4 weeks and agrees with the two months observed in other reports [31]—did not differ between the short and long CMV-Ig schedule. Secondly, it was similar between those who prematurely discontinued VGCV and those who did not, and did not vary between CMV-Ig schedules among the subgroup of recipients who completed the recommended 12 months of VGCV prophylaxis. Thirdly, premature antiviral prophylaxis discontinuation was the only independent risk factor for time to first CMV infection in the multivariate analysis, while the CMV-Ig schedule had no effect. Lastly, the survival analysis showed that the cumulative risk of the first CMV infection off antiviral prophylaxis did not differ between the extended or short CMV-Ig schedules.

Overall, 24% of patients discontinued VGCV prophylaxis in our study for reasons other than CMV breakthrough (mainly myelotoxicity), which agrees with the 23.6% observed in a previous report in D+/R–lung recipients receiving only VGCV prophylaxis [32] Lung transplant and D+/R– mismatch are known independent risk factors for premature VGCV cessation [32]. Moreover, the incidence of CMV viremia following antiviral discontinuation is higher among D+/R–lung recipients than other serologies in patients receiving a short course (7 doses) of adjunctive CMV-Ig [33]. Additionally, VGCV duration <6 months is independently associated with CMV disease in D+/R–lung recipients [31]. Thus, our study’s close association between VGCV cessation and a shorter time to first CMV infection in patients treated with the ER-Ig regimen is not surprising, although the reasons behind the higher proportion of interruptions in this group are unclear. As therapeutic drug monitoring is not routinely indicated [34], we cannot discard different proportions of VGC/VGCV toxicity between CMV-Ig schedule groups. However, this is not probable considering that all centers used the same GCV/VGCV dosage. A more plausible explanation would involve different management strategies between centers when toxicity emerged: The usual tendency for ER-Ig centers is to discontinue VGCV early when side effects appear, probably influenced by CMV-Ig coverage up to one year after transplantation, while the SR-Ig center has a tendency to withdraw or reduce doses of other drugs when myelotoxicity appears (such as mycophenolate mofetil or trimethoprim-sulfamethoxazole), or reduce doses of VGCV adjusted for renal function, but without suspending it.

At present, other antiviral treatments against CMV have proven to be useful in prophylaxis with fewer side effects. Letermovir, a drug approved for CMV-prophylaxis of bone marrow transplantation, has been shown to be beneficial in lung transplantation with fewer adverse effects in some case series [35,36,37,38,39,40]. Recently, Maribavir has been shown in a phase 3 clinical trial to be useful in infections refractory to conventional treatment, with less neutropenia and less renal failure [41]. In the future, these drugs could be helpful in managing side effects in high-risk CMV-matched lung transplantation

One striking finding in our study was the identification of PAH as the only significant risk factor for the time to first CMV infection after antiviral prophylaxis discontinuation. The explanations for this observation are not straightforward and might be related to the pro-inflammatory environment associated with PAH [42,43]. Indeed, a significant adaptative immune system dysregulation has been reported in PAH patients, with decreased levels of CD4+ and CD8+ T lymphocytes and natural killer (NK) cells and reduced cytokine-producing capacity in peripheral blood [43,44]. Interestingly, a recent study identified a weak correlation between Epstein-Barr virus (EBV) loads and overexpression of PD-1 (programmed cell death protein 1), which is involved in the inhibition of lymphocyte activation, in patients with PAH [38]. The authors concluded that the immune system dysregulation in PAH may contribute to increased susceptibility to EBV reactivation and also de novo infection [45]. Considering that both CMV and EBV are human herpesviruses that share common structural characteristics, life cycle (i.e., cellular invasion, replication, and latency), and similar innate and adaptive (particularly CD8+ T cells) responses of the host upon infection [46,47], it is tempting to speculate that PAH could also increase the risk of CMV reactivation once the VGCV prophylaxis was discontinued.

The main advantage of the study is that it was a real-world, non-interventional analysis to determine the optimal duration of two different CMV-Ig adjunctive prophylaxis in lung transplant recipients. However, the study has several limitations that must be acknowledged. Firstly, the sample size could be considered small, and the statistical power not optimal to claim for validity and generalizability of the results. This cohort represents approximately 35% of all lung transplantations conducted between 2009 and 2020 in three out of the seven accredited centers in Spain [48]. Therefore, the results need to be confirmed by independent researchers using CMV-Ig as part of the universal CMV prophylaxis in large multicenter studies. Secondly, as a non-randomized trial, we cannot completely discard imbalances between the two treatment cohorts despite confounding effects being considered in the statistical methods. Lastly, the retrospective design using data extracted from electronic medical records has inherent limitations because it relies on the quality of record keeping, thus susceptible to incomplete, inaccurate, or missing values. 

## 5. Conclusions

Our preliminary results add to the literature that extending the adjunctive CMV-Ig prophylaxis beyond the manufacturer’s recommendations (i.e., at least 6 single doses over approximately 2.5 months) up to one year does not confer additional benefits regarding post-transplantation outcomes in lung transplant CMV mismatched recipients. Based on the similar clinical benefit of the two approaches, we propose to favor short CMG-Ig schedules over extended regimens because IV infusion is generally inconvenient for the health provider and the patient, and a longer than the strictly necessary duration of the CMV-Ig therapy entails increased pharmacy costs, added logistical challenges (with more doses given on an outpatient basis), and a negative impact in the patient’s quality of life.

## Figures and Tables

**Figure 1 microorganisms-11-00032-f001:**
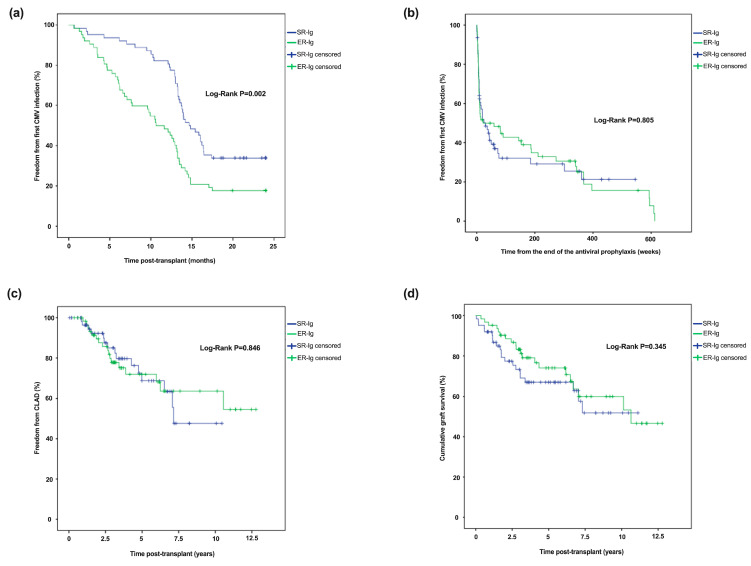
Kaplan-Meyer analyses of (**a**) time first CMV infection; (**b**) cumulative risk of first CMV infection off antiviral prophylaxis; (**c**) time to CLAD; and (**d**) survival in patients receiving short or label use CMV-Ig prophylaxis (SR-Ig) compared to patients receiving extended CMV-Ig regimen (ER-Ig).

**Table 1 microorganisms-11-00032-t001:** Baseline demographic and clinical characteristics, immunosuppressive strategies, and CMV-prophylaxis of the recipients included in the study by adjunctive CMV-Ig schedule received.

	All Recipients (*n* = 124)	CMV-Ig Short Regimen (*n* = 62)	CMV-Ig Extended Regimen (*n* = 62)	*p*-Value
Number of centres, *n*	3	1	2	
Age *, years, median (IQR)	51.6 (38.0–60.8)	56.1 (43.0–62.1)	47.4 (34.3–57.3)	0.013
Sex, male, *n* (%)	74 (59.7)	37 (59.7)	37 (59.7)	1.000
Bilateral lung transplantation, *n* (%)	103 (83.1)	51 (82.3)	52 (83.9)	0.811
Indication for lung transplant, *n* (%)				0.428
COPD	29 (23.4)	14 (22.6)	15 (24.2)	
ILD	40 (32.3)	25 (40.3)	15 (24.2)	
Bronchiectasis	30 (24.2)	12 (19.4)	18 (29.0)	
PAH	13 (10.5)	6 (9.7)	7 (11.3)	
Lung retransplantation	1 (0.8)	0 (0)	1 (1.6)	
Other	11 (8.9)	5 (8.1)	6 (9.7)	
Induction therapy **, *n* (%)	101 (81.5)	39 (62.9)	62 (100)	<0.001
Maintenance therapy, *n* (%)				
Calcineurin inhibitors				0.619
Tacrolimus	120 (96.8)	59 (95.2)	61 (98.4)	
Cyclosporine	4 (3.2)	3 (4.8)	1 (1.6)	
Anti-metabolites				<0.001
MMF/MPS	115 (92.7)	62 (100)	53 (85.5)	
Azathioprine	9 (7.3)	0 (0)	9 (14.5)	
Corticoids	123 (99.2)	62 (100)	61 (98.4)	0.315
Azithromycin	86 (69.4)	47 (75.8)	39 (62.9)	0.172
mTOR inhibitor	37 (29.8)	6 (9.7)	31 (50.0)	<0.001
CMV antiviral prophylaxis, *n* (%)				
GCV/VGCV	100	100	100	1.000
Other (leflunomide)	2 (1.6)	0 (0)	2 (3.2)	0.496
Follow-up, years, median (IQR)	3.65 (2.0–6.9)	3.50 (1.5–6.7)	4.10 (2.9–7.1)	0.236

* Age at the transplant. ** The drug used was, in all cases, basiliximab. CMV, cytomegalovirus; COPD, chronic obstructive pulmonary disease; ILD, interstitial lung disease; GCV, ganciclovir; IQR, interquartile range; MMF, mycophenolate mofetil; MPA; mycophenolic acid; PAH, pulmonary arterial hypertension; SD, standard deviation; VGCV, valganciclovir.

**Table 2 microorganisms-11-00032-t002:** Post-transplant outcomes of the lung recipients included in the study by adjunctive CMV-Ig regimen received.

	All Recipients (*n* = 124)	CMV-Ig Short Regimen (*n* = 62)	CMV-Ig Extended Regimen (*n* = 62)	*p*-Value
VGCV duration, months, median (IQR)	12 (10–12)	12 (12–13)	12 (7–12)	0.003
Premature VGCV prophylaxis discontinuation, *n* (%)	31 (25)	5 (8.1)	26 (41.9)	<0.001
VGCV duration, months, median (IQR) among those who discontinued	7 (5–10)	10 (7–12)	7 (4.8–10)	0.291
Indication for discontinuation, *n* (%)				
Myelotoxicity	13 (41.9)	1 (20.0)	12 (46.9)	0.284
Renal failure	8 (25.8)	1 (20.0)	7 (26.9)	
CMV infection	7 (22.6)	3 (60.0)	4 (15.4)	
Gastrointestinal distress	2 (6.5)	0 (0.0)	2 (7.7)	
Non-adherence	1 (3.2)	0 (0.0)	1 (3.8)	
CMV infection, *n* (%)	92 (74)	42 (67.7)	50 (80.6)	0.150
First CMV infection				
Time from transplant, months, median (IQR)	13.30 (7.81–18.33)	14.75 (12.93–21.28)	11.13 (5.56–14.65)	<0.001
Time from the end of antiviral CMV prophylaxis, weeks, median (IQR)	7.43 (4.43–14.0)	8.57 (5.03–20.60)	6.85 (4.21–10.28)	0.209
Tissue-invasive disease, *n* (%)	11 (12)	5 (11.9)	6 (12.0)	0.989
Pneumonitis	4 (36.4)	2 (40)	2 (33.3)	0.569
Gastritis	2 (18.2)	0 (0.0)	2 (33.3)	
Colitis	2 (18.2)	1 (20.0)	1 (16.7)	
Hepatitis	2 (18.2)	1 (20.0)	1 (16.7)	
Colitis and hepatitis	1 (9.1)	1 (20.0)	0 (0.0)	
Number of CMV infections, median (IQR)	2 (1–3)	1 (1–2)	2 (1–3)	<0.001
With IV treatment	1 (1–1)	1 (1–1)	1 (1–1.25)	0.325
With oral treatment	1 (1–2)	1 (1–2)	1 (1–2)	0.091
Acute rejections, *n*, median (IQR)	1 (1–2)	1 (1–2)	1.5 (1–2)	0.902
Severity grade, *n*, median (IQR)				
A1-A2	1 (1–2)	1 (1–2)	1 (1–2)	0.919
A23-A4	1 (1–1.25)	1 (1–2)	1 (1–1)	0.454
Treated with corticosteroids megadose, *n* (IQR)	1 (1–2)	1 (1–2)	1 (1–2)	0.830
Treated with oral corticosteroids, *n* (IQR)	1 (1–1)	1 (1–1)	1 (1–1)	1.000
CLAD, *n* (%)	32 (25.8)	15 (24.2)	17 (27.4)	0.838
BOS	29 (90.6)	14 (93.3)	15 (88.2)	0.621
RAS	3 (9.4)	1 (6.7)	2 (11.8)	
Time from transplant, years, median (IQR)	3.16 (1.70–6.15)	3.18 (1.38–5.68)	3.15 (2.01–6.21)	0.055
Death, *n* (%)	41 (33.1)	21 (33.9)	20 (32.2)	0.849

BOS, bronchiolitis obliterans syndrome; CLAD, chronic lung allograft dysfunction; CMV, cytomegalovirus; IQR, interquartile range; IV, intravenous; RAS, restrictive allograft syndrome; SD, standard deviation; VGCV, valganciclovir.

**Table 3 microorganisms-11-00032-t003:** Risk factors for first CMV infection using logistic regression.

Variables	Crude OR (95% CI)	*p*-Value	Adjusted OR (95% CI)	*p*-Value
Age at transplant	0.964 (0.933–0.997)	0.034	-	-
Months of VGCV prophylaxis	1.039 (0.944–1.144)	0.436		
Type of transplant Unilateral Bilateral	0.383 (0.144–1.022) Ref.	0.055		
Immunosuppression induction Yes No	1.711 (0.647–4.526) Ref.	0.279		
Indication for lung transplant COPD DILD Bronchiectasis PAH Other	Ref. 0.791 (0.277–2.259) 1.048 (0.332–3.302) 4.571 (0.508–41.114) 3.810 (0.417–34.763)	0.662 0.937 0.175 0.236		
Acute rejection	0.999 (0.591–1.690)	0.997		
Acute rejection severity A1-A2 A3-A4	0.607 0.050	0.165 0.063		
Acute rejection treated with corticosteroids megadose	0.974 (0.555–1.710)	0.926		
Premature VGCV prophylaxis discontinuation Yes No	4.229 (1.189–15.043) Ref.	0.026	4.083 (1.146–14.552)	0.030
CMV-Ig schedule SR-Ig ER-Ig	Ref. 1.984 (0.870–4.527)	0.104		

CMV, cytomegalovirus; COPD, chronic obstructive pulmonary disease; ER-Ig, extended CMV-Ig; DILD, diffuse interstitial lung disease; GVC, ganciclovir; SR-Ig, label use or short CMV-Ig regimen; OR, odds ration; PAH, pulmonary arterial hypertension; VGCV, valganciclovir.

## Data Availability

The data are available from the corresponding author, upon reasonable request.

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
