# Peer review of "Evaluation of Two Different CMV-Immunoglobulin Regimens for Combined CMV Prophylaxis in High-Risk Patients following Lung Transplant"

_microorganisms, 2022, doi:10.3390/microorganisms11010032_

Round 1

Reviewer 1 Report

Microorganisms 2098132

CMV is the most important pathogen in transplant recipients because of opportunistic infection.  As the infection of CMV in transplant recipients leads to severe complications, it is necessary to prevent the replication and reactivation of virus using with antiviral drugs and other treatments including immunoglobulin therapy.  In this report, the authors compared the clinical outcome between label and off-labeled use of CMV-Ig treatment (2.5 months and over one year after transplantation, respectively) as retrospectively.  However, the extension of adjunctive CMV-Ig prophylaxis did not present additional clinical benefits.  These results were important to prevent the over-treatment and meaningful to improve the QOL of patients.

Specific comment

As shown in Table 2, ER-Ig group presented shorter VGCV treatment duration.  As described in line 217, adverse effects were the most frequent reasons for premature discontinuation.  However, it is unclear why these adverse effects occurred more frequently in ER-Ig group.  Did basiliximab, used as induction therapy and treated to all ER-Ig patients, possess any negative effects in this group?  In addition, several reports presented that the beneficial effects mTOR inhibitors to prevent the replication of CMV.  In this report, many patients were treated mTORi in ER-Ig group (Table 1, line 202).  The authors should describe the effects of these adjunctive treatments.

Minor comments

Table 1, the 4th row, “Bilateral lung transplantation, n (%)”

              83.1%, 82.3%, 83.9%

Table 1, the 16th row, “Cyclosporine”

              3.2, 4.8%, 1.6

Table 2, the second row, “Premature VGCV prophylaxis discontinuation, n(%)”

25, 8.1%, 41.9

Author Response

Thanks for your comments. See attached document.

Reviewer 2 Report

This is a very well designed study, taking advantage of access to similar populations treated with two different regimens of Ig to control CMV in lung transplant patients. The results are clear and of high interest for guiding clinical practice in a field that is characterized by heterogeneous regimens. The Discussion could benefit from noting that letermovir has been approved (for bone marrow transplant) since the study ended and it is likely to reduce the fraction of patients who terminate antiviral treatment due to toxicity in solid organ transplant. However, that should not affect the conclusions of the study regarding use of CMV-Ig. 

Author Response

(The authors gave the same response as above.)
